# Neural-based classification rule learning for sequential data

**Marine Collery**[1,2]*, **Philippe Bonnard**[1], **François Fages**[2] **& Remy Kusters**[1,3]
[1]IBM France Lab, [2]Inria Saclay, [3]IBM Research

## Abstract

Discovering interpretable patterns for classification of sequential data is of key importance for a variety of fields, ranging from genomics to fraud detection or more generally interpretable decision-making. In this paper, we propose a novel differentiable fully interpretable method to discover both local and global patterns (i.e. catching a relative or absolute temporal dependency) for rule-based binary classification. It consists of a convolutional binary neural network with an interpretable neural filter and a training strategy based on dynamically-enforced sparsity. We demonstrate the validity and usefulness of the approach on synthetic datasets and on an open-source peptides dataset. Key to this end-to-end differentiable method is that the expressive patterns used in the rules are learned alongside the rules themselves.

## 1 Introduction

During the last decades, machine learning and in particular neural networks have made tremendous progress on classification tasks for a variety of fields such as healthcare, fraud detection or entertainment. They are able to learn from various data types ranging from images to timeseries and achieve impressive classification accuracy. However, they are difficult or impossible to understand by a human. Recently, explaining those black-box models has attracted considerable research interest under the field of Explainable AI (XAI). However, as stated by Rudin (2019), those aposteriori approaches are not the solution for high stakes decision-making and more interest should be placed on learning models that are interpretable in the first place.

Rule-based methods are interpretable, human-readable and have been widely adopted in different industrial fields with Business Rule Management Systems (BRMS). In practice however, those rules are manually written by experts. One of the reasons manually-written rule models cannot easily be replaced with learned rule models is that rule-base learning models are not able to learn as expressive rules with higher-level concepts and complex grammar (Kramer, 2020). Moreover, due to the lack of latent representations, rule-based learning methods underperform w.r.t. state-of-the-art neural networks (Beck & Fürnkranz, 2021).

Classical classification rule learning algorithms (Cohen, 1995; Breiman et al., 1984; Dash et al., 2018; Lakkaraju et al., 2016; Su et al., 2016) as well as neural-based approaches to learn rules (Qiao et al., 2021; Kusters et al., 2022) (or logical expressions with Riegel et al. (2020)) do not provide the grammar required to learn classification rules on sequential data. Numerous approaches for learning classification rules on sequential data in the field of sequential pattern mining have been studied in the past such as Egho et al. (2015); Zhou et al. (2013); Holat et al. (2014) but with a different goal in mind : improve the performance of extracted patterns for a fixed rule grammar as opposed to extending the rule grammar. Another domain of research focuses on training binary neural networks to obtain more computational efficient model storing, computation and evaluation efficiency (Geiger & Team, 2020; Helwegen et al., 2019). It comes with fundamental optimization challenges around weights updates and gradient computation.

In this paper, we bridge three domains and introduce a binary neural network to learn classification rules on sequential data. We propose a differentiable rule-based classification model for sequential data where the conditions are composed of sequence-dependent patterns that are discovered

---

*Corresponding author `marine.collery@ibm.com`

*alongside* the classification task itself. More precisely, we aim at learning a rule of the following structure: *if* pattern *then class* = 1 *else class* = 0. In particular we consider two types of patterns: local and global patterns as introduced in Aggarwal (2002) that are in practice studied independently with a local and a global model. A local pattern describes a subsequence at a specific position in the sequence while a global pattern is invariant to the location in the sequence (Fig 2). The network, that we refer to as Convolutional Rule Neural Network (CR2N), builds on top of a *base rule model* that is comparable to rule models for tabular data presented in Qiao et al. (2021); Kusters et al. (2022).

The contributions of this paper are the following: i) We propose a convolutional binary neural network that learns classification rules together with the sequence-dependent patterns in use. ii) We present a training strategy to train a binarized neural network while dynamically enforcing sparsity. iii) We show on synthetic and real world datasets the usefulness of our architecture with the importance of the rule grammar and the validity of our training process with the importance of sparsity. The code is publicly available at https://github.com/IBM/cr2n.

## 2 BASE RULE MODEL

The base rule model we invoke is composed of three consecutive layers (Fig 1). The two last layers respectively mimic logical AND and OR operators (Qiao et al., 2021; Kusters et al., 2022). On top of these layers, we add an additional layer that is specific for categorical input data and corresponds to an OR operator for each categorical variable over every possible value it can take.

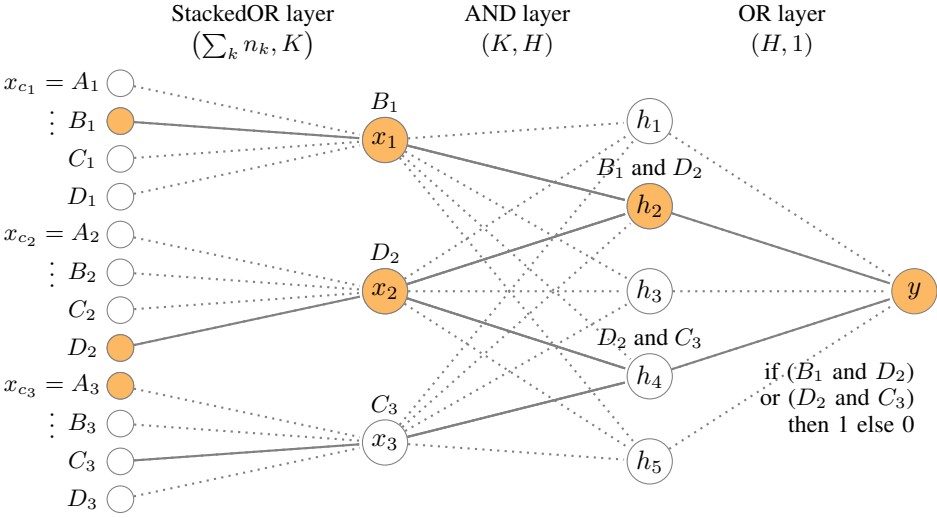

Figure 1: Example of a trained *base rule model* architecture for the rule *if ($B_1$ and $D_2$) or ($D_2$ and $C_3$) then 1 else 0* on 3 categorical variables $x_{c_1}$, $x_{c_2}$ and $x_{c_3}$ ($x_{c_k} \in \{A_k, B_k, C_k, D_k\}$). For simplicity, the truth value of $x_{c_1} = B_1$ is replaced by $B_1$ for example. Plain (dotted) lines represent activated (masked) weights. An example evaluation of the model is represented with the filled neurons (neuron=1) for the binary input $x_{c_1} = B_1$, $x_{c_2} = D_2$ and $x_{c_3} = A_3$.

The AND layer takes binary features (which are atomic boolean formulae) as input and outputs to the OR layer. The output of the OR layer is mapped to the classification label $y$. These layers have binary weights specifying the nodes that are included in the respective boolean expression (conjunction or disjunction). In other words, this network implements the evaluation of a DNF and has a direct equivalence with a binary classification rule like *if $(A \land B) \lor C$ then class* = 1 *else class* = 0, where $A$, $B$ and $C$ are binary input features (atoms in logical terms). In this paper, we focus on supervised binary classification where we predict the label $y \in \{0, 1\}$ given input data $\boldsymbol{x}$.

The base rule model is illustrated in Fig 1 and is composed of three binary neural layers.

- Input neurons $\boldsymbol{x}$, are binarized input features of size $K$ ($\boldsymbol{x}_c$ are one-hot encoded categorical input features of size $\boldsymbol{n}$).

- Hidden neurons $\boldsymbol{h}$, are conjunctions of the input features of size $H$.
- Output neuron $\boldsymbol{y}$, is a disjunction of the (hidden) conjunctions.

We assign to each of boolean operations, i.e. AND and OR operations, a binary weight ($\boldsymbol{W}_{and}$ and $\boldsymbol{W}_{or}$ respectively) that plays the role of a mask to filter nodes with regards to their respective logical operation. For the sake of simplicity, we did not extend the model with a logical NOT operation.

The disjunction operation is implemented as,

$$y = \min(\boldsymbol{W}_{or}\boldsymbol{h}, 1). \tag{1}$$

If none of the neurons $h$ are activated then $y = 0$, and $y = 1$ if at least one is.

For the conjunction operation, we use the De Morgan's law that express the conjunction with the OR operator $A \wedge B = \neg(\neg A \vee \neg B)$. Combined with Eq 1, we obtain:

$$\boldsymbol{h} = \neg(\min(\boldsymbol{W}_{and}(\neg\boldsymbol{x}), \mathbf{1})) = \mathbf{1} - \min(\boldsymbol{W}_{and}(\mathbf{1} - \boldsymbol{x}), \mathbf{1}). \tag{2}$$

**StackedOR Input Layer** As defined previously, the AND layer takes binary input features as input. In this paper, we propose to add an additional layer for categorical data. A categorical variable $\boldsymbol{x}_c$ can take one value $\alpha_i^c$ out of a fixed number of possible values $n$ e.g. $\{\alpha_0, \ldots, \alpha_3\} = \{A, B, C, D\}$. Without any additional layer, it requires a one-hot encoding to be provided as input to the AND layer. Binary inputs $x_c = A$ and $x_c = B$ are then given as input to the AND layer that can in theory represent the impossible expression $x_c = A \wedge x_c = B$ i.e the model has to learn the hidden categorical relationship between the one-hot encoded variables. To prevent learning a distribution we already know, we deepen the model with the addition of a stacked architecture of OR layers as input of the AND layer as shown in Fig 1. This structure is defined by $K$ weights, $\boldsymbol{W}_{stack}^k$, for each input categorical variables and will be referred to as the StackedOR layer with $\boldsymbol{W}_{stack}$ weights.

To conclude, the base rule model is composed of a StackedOR layer for categorical variables, a logical AND layer and an OR layer. The formal grammar that this architecture can express is specified with the following production rules (see Appendix A for the full grammar):

$$
\begin{aligned}
\text{rule} &\rightarrow \underline{\text{if}} \text{ base expression}\underline{\text{then class}} = 1 \underline{\text{else class}} = 0 \\
\text{base expression} &\rightarrow \text{conjunction} \mid \text{conjunction} \vee \text{base expression} \\
\text{conjunction} &\rightarrow \text{predicate} \mid \text{predicate} \wedge \text{conjunction} \\
\text{predicate} &\rightarrow \text{categorical expression} \mid \text{literal} \\
\text{categorical expression} &\rightarrow \text{categorical literal} \mid \text{categorical literal} \vee \text{categorical expression} \\
\text{categorical literal} &\rightarrow \underline{x_c = \alpha_0^c} \mid \ldots \mid \underline{x_c = \alpha_{n_c}^c} \text{ (or simply } \underline{\alpha_0^c} \mid \ldots \mid \underline{\alpha_{n_c}^c}) \\
\text{literal} &\rightarrow \underline{x_1} \mid \underline{x_2} \mid \ldots
\end{aligned}
\tag{3}
$$

This grammar is also limited by the model architecture: *conjunction* contains at most one occurrence of each *predicate* and the total number of *conjunction*(s) is bounded by the number of hidden nodes.

## 3 Convolutional Rule Neural Network

Our main contribution is to extend the base rule model for sequential data. We apply the base rule model as a 1D-convolutional window of fixed length $l \in \mathbb{N}$ over a sequence and retrieve all outputs as input for an additional disjunctive layer which we refer to as the ConvOR layer as shown in Fig 2 [1]. The base rule model learns a DNF over the window size length and the ConvOR layer indicates where along the sequence that logical expression is true. If the evaluation of the logical expression is true all along the sequence then it can be described as a *global pattern*, otherwise the learned pattern represents a *local pattern*.

The model input is now of size $\sum_k l \times n_k$ and output of StackedOR layer (or input of the AND layer) is $l \times K$. Other dimensions are not impacted. For simplicity in the following, $K$ is fixed to 1 i.e. input data is composed of one categorical variable evolving sequentially. The method is still valid for $K > 1$. Fig 1 is also still valid with a change of index, $k$ is now referring to the position in the window of size $l$.

---

[1] A natural extension for sequential data of the base rule architecture would be to extend it with an explicit recursion of the base rule model, similar to a RNN. This approach was tested but faced the same limitations as any classical RNNs, i.e., vanishing gradients and only captures short-term dependencies.

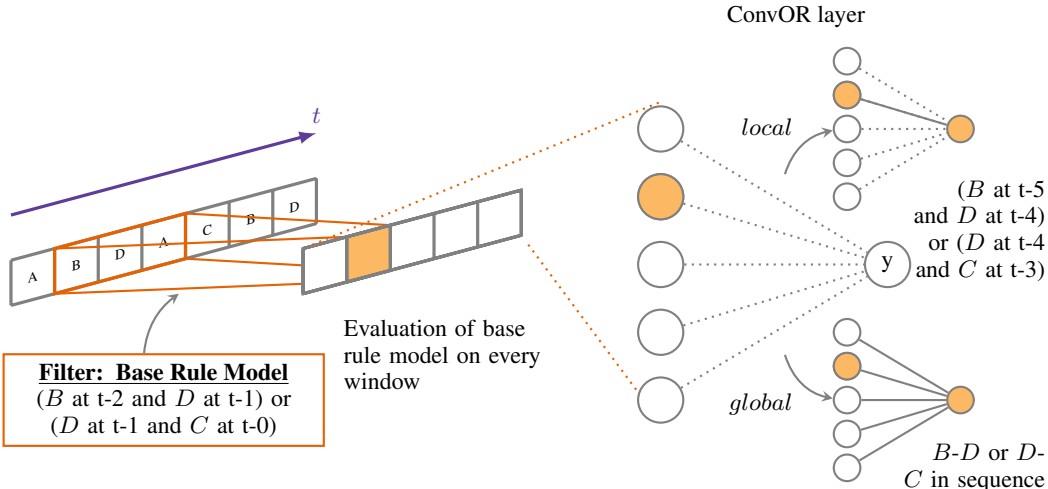

Figure 2: Example of a trained CR2N architecture. The base rule model is applied as 1D-convolutional window over the sequence (i.e. sliding window). The resulting boolean values are given as input of the ConvOR layer which indicates through its activated weights where along the sequence the expression learned by the base model is true. The output of the ConvOR layer is mapped to the label of the sequence $y$. For local patterns, the base model expression needs to be shifted accordingly to the ConvOR layer weights. For a real-domain application like fraud detection, by providing meaning to B, C and D, we could have for example if "receiving a transaction of amount X"($B$) is followed by "emitting a transaction of amount X" ($D$) or "emitting a transaction of amount X"($D$) is followed by "closing the bank account"($C$) then class=fraud.

With this approach, different sequence-dependent expressions can be extracted and their nature depends on the weights of the ConvOR layer (Fig 2). If all the weights, $W_{conv}$, of the ConvOR layer are activated (i.e. equal to 1), the logical expression learned by the base model is true in all the sequence: a *global* pattern is learned. If only some of the weight of the ConvOR layer are activated, the logical expression learned by the base model is valid only in the window associated to that weight: a *local* pattern is learned. The base model logical expression is modified accordingly to match that shift (see example in Fig 2 with a shift of 3 sequential steps).

The obtained weights thus translate to a rule grammar with the following production rules:

$$
\begin{aligned}
\text{rule} &\rightarrow \textit{if} \text{ expression } \textit{then class} = 1 \textit{ else class} = 0 \\
\text{expression} &\rightarrow \text{local pattern} \mid \text{global pattern}
\end{aligned} \tag{4}
$$

We introduce $t$, the position when the last observation in a sequence was made. With $t$ being our reference, in a sequence of size $N \in \mathbb{N}$, $t - i$ refers to the moment of the $i^{\text{th}}$ observation before $t$ ($0 \leq i \leq N - 1$). $A, B, C$ and $D$ are toy binary input possible values for our categorical variable $x_c$ (they cannot be activated simultaneously at the same position $t$ in the sequence). With those definitions, we list below examples of different sequence-dependent expressions that can be expressed with the proposed architecture (see Fig 2):

A **local pattern** is an expression composed of predicates that are true at a specific position $i$, for example *A at t-15*. Based on Eq 3 we have:

$$
\begin{aligned}
\text{local pattern} &\rightarrow \text{base expression} \\
\text{predicate} &\rightarrow \text{categorical expression } \textit{at t-i} \mid \text{literal } \textit{at t-i.}
\end{aligned} \tag{5}
$$

A **global pattern** is an expression describing the presence of a pattern anywhere in the sequence, for example *B-D in sequence* is a global pattern where "$-$" sign refers to "followed by" and "$*$" correspond to any unique literal (equivalent to $\forall i \in [0; N-1]$, *B at t-i-1 and D at t-i*) . If inputs are sequences of characters, global patterns can be compared to simple regular expressions supporting the logical OR (metacharacter'[ ]'). Based on Eq 3 we have:

$$
\begin{aligned}
\text{global pattern} &\rightarrow \text{base expression} \\
\text{conjunction} &\rightarrow \text{predicate} \mid * \mid \text{predicate} - \text{conjunction}
\end{aligned} \tag{6}
$$

Additional special cases can be pointed out such as the learning of a global pattern over an interval (e.g. *B-\*-D in window [t-6; t-3]*) or the learning of sequence characteristics dependant expression such as *4 ≤ len(sequence) ≤ 6* based on the sequence length (not shown on Fig 2 but it corresponds to a specific case where the base model has learned an always true rule). Also, it is important to note that *base expression* and *conjunction* in both grammars are bounded by the fixed window size $l$.

To ensure full equivalence between the model and rule, sequences boundaries need to be considered, especially for global patterns. All sequences are padded on both ends with a sequence of 0 of size $l - 1$ (not shown for simplicity on Fig 2). Also sequences of different lengths are supported by creating a model based on the maximal available sequence length $M$ in the data and padding shorter sequences with a sequence of 0 of appropriate length. ConvOR layer input size is then $M + l - 1$.

With this one architecture we can model both local and global patterns. However for optimization reasons detailed below, we choose to differentiate the two into two distinct models: a **local** and a **global model**. The ConvOR layer weights for the global model are set and fixed to 1 during training.

## 4 TRAINING STRATEGY

To overcome training challenges attributed to binarized neural networks (Geiger & Team, 2020), we use latent weights and enforce sparsity dynamically. We define a loss function that penalizes complex rules and the model is trained via automatic differentiation (Pytorch) with Adam optimizer.

**Latent weights**   The binary model parameters introduced above ($\boldsymbol{W}_{and}$, $\boldsymbol{W}_{or}$, $\boldsymbol{W}_{stack}$, $\boldsymbol{W}_{conv}$) are trained indirectly via the training of a continuous parameter $loc$ which is activated (binarized) by a sigmoid function (Kusters et al., 2022). With such binary weights and continuous relaxation Eq 1 and 2 are differentiable with nonzero derivatives (Kusters et al., 2022). As opposed to when using a straight through estimator (Qiao et al., 2021), non-zero gradients are ensured during the backward pass. To overcome training limitations, we use a hard concrete distribution (Qiao et al., 2021; Louizos et al., 2018). It rescales the weights and the random variable introduced during training prevents from obtaining local minima (Appendix B). Weight values are in $[0, 1]$ during training, while for testing and rule extraction, a Heaviside is applied to them ($\geq 0.5$) to ensure strict binarization.

**Loss function**   We define the loss function $\mathcal{L}$ composed of a mean-squared error component along with a regularization term that penalize the complexity of the rule,

$$\mathcal{L} = \mathcal{L}_{mse} + \lambda \Pi \tag{7}$$

That regularization term $\Pi$, or *penalty*, evaluates the number of terminal conditions in the rule. In practice we use $\lambda = 10^{-5}$. For a $\text{layer}_n$ of input size $I$ and output size $O$, the number of terminal conditions per output corresponds to the weighted sum of the number of terminal conditions of each output of the previous layer i.e. $\boldsymbol{\Pi}_{\text{layer}_n} = \sum_I \boldsymbol{W}_{\text{layer}_n} \boldsymbol{\Pi}_{\text{layer}_{n-1}}$, a vector of size $O$. For the first layer, the StackedOR layer, $\boldsymbol{\Pi}_{stack}$ is defined as the sum over the input dimension of the weights and we can then express the number of terminal conditions of the base rule model $\Pi_{base}$.

$$\boldsymbol{\Pi}_{\text{layer}_0} = \boldsymbol{\Pi}_{stack} = \begin{bmatrix} \sum \boldsymbol{W}_{stack}^1 \\ \vdots \\ \sum \boldsymbol{W}_{stack}^K \end{bmatrix}, \qquad \Pi_{base} = \sum \boldsymbol{W}_{or} \sum \boldsymbol{W}_{and} \boldsymbol{\Pi}_{stack} \tag{8}$$

For optimizing $\Pi$ for local patterns, we have to minimize the activated ConvOR layer weights. For global patterns, we want them to all be activated. A condition could be set on the sum of ConvOR layer weights (Eq 9) to shift from one optimization problem to the other but with loss of continuity and thus differentiability (interesting values of $\tau$ being $M + l - 1$, the ConvOR layer input size, that would correspond to all ConvOR layer weights being equal to 1, or $M - l + 1$ that would allow for $2(l - 1)$ weights to be 0, and corresponds to the padding required for properly accounting for sequence boundaries (Section 3)).

$$\Pi_{local} = \Pi_{base} \sum \boldsymbol{W}_{conv}, \qquad \Pi_{global} = \Pi_{base}, \qquad \Pi^* = \begin{cases} \Pi_{global} & \text{if } \sum \boldsymbol{W}_{conv} \geq \tau \\ \Pi_{local} & \text{otherwise} \end{cases} \tag{9}$$

Table 1: Ground truth applied on sequences of letters (A to F) to generate synthetic unbalanced datasets 1, 2, 3 and 4 along with the distributions of the positive class. In the patterns, $t$ refers to the position when the last observation in a sequence was made. Balanced datasets with same ground truth are generated and are referred to as the dataset number followed by the letter $b$ (Appendix D).

| Ground truth | # | Distribution (%) |
|---|---|---|
| C at t-4 | 1 | 14.2 |
| A at t-6 and C at t-4 | 2 | 1.5 |
| (A at t-6 and C at t-4) or (B at t-5 and C at t-3) | 3 | 3.6 |
| B-D in sequence | 4 | 20.4 |

Due to non continuity of $\Pi^*$ in Eq 9, we choose to have two models with the same architecture for the two cases: the *local* and the *global model* respectively more relevant for their associated pattern. For the local model, all weights are trainable and $\Pi = \Pi_{local}$. For the global model, weights in the ConvOR layer are fixed and set to 1, and $\Pi = \Pi_{global}$.

**Enforced sparsity** Sparsity of the model is crucial to learn concise expressions, the model needs to generalize without observing *all* possible instances at training time. The first requirement for that matter is sparsity in the base rule model. In addition to the regularization term in the loss function, we propose to use a sparsify-during-training method (Hoefler et al., 2021) and dynamically enforce sparsity in weights from 0% to an end rate $r_f$ set to 99% in our case (Lin et al., 2020). Sparsify-during-training method can also benefit the quality of the training in terms of convergence by correcting for approximation errors due to premature pruning in early iterations but is highly dependant on the sparsification schedule (Hoefler et al., 2021).

Every 16 iterations $s$ and for a total of $s_f$ training iterations, every trainable weight is pruned with a binary mask, $\mathbf{m}$, (of size of its associated weight and applied with Hadamard product ($\odot$)) (Lin et al., 2020; Zhu & Gupta, 2017). We propose a mask based on the maximum of weight magnitude $loc$ and pruning rate $r$ (Zhu & Gupta, 2017) making the assumption that it contributes to generalization (Eq 10). This strategy can be more aggressive than state-of-art contributions (Lin et al., 2020) due to its dependency to the $loc$ maximum value. During training, the model with the highest prediction accuracy on validation dataset and the highest sparsity (evaluated at each epoch) is kept.

$$r = r_f - r_f \left( 1 - \frac{s}{s_f} \right)^3, \qquad m_{i,j} = |loc_{i,j}| \geq r \times \max_{i,j}(loc), \qquad \hat{\mathbf{W}} = \mathbf{W} \odot \mathbf{m} \qquad (10)$$

Additional training optimizations have been tested out such as for example using a binarized optimizer (Helwegen et al., 2019; Geiger & Team, 2020), adding a scheduled cooling on the sigmoid of the binarized weights, alternating the training of each layer every few epochs (Qiao et al., 2021) or using a learning rate scheduler. Those techniques are not presented here but would be of interest for improving results on specific datasets.

## 5 EXPERIMENTS

In order to evaluate the validity and usefulness of this method, we apply it to both synthetic datasets and UCI membranolytic anticancer peptides dataset (Grisoni et al., 2019; Dua & Graff, 2017).

**Synthetic Datasets** We propose 8 synthetic datasets based on 4 ground truth expressions in both balanced and unbalanced distributions for discovering simple binary classification rules with local or global patterns as shown in Table 1. There are 1000 sequences of letters (A to F) of different length from 4 to 14 letters in each of them (Mean around 9±3). Generation is detailed in Appendix D.

**Peptides Dataset** Besides the synthetic datasets, real-world UCI anticancer peptides dataset composed of labeled one-letter amino acid sequences, is used (Grisoni et al., 2019; Dua & Graff, 2017). The multi-classification problem is transformed into a binary classification problem is the same manner as Nwegbu et al. (2022) (see Appendix C). Sequence length are from 5 to 38 letters (Mean: $17 \pm 5.5$) and positive class distribution is 79%.

**Experimental Setting** All datasets are partitioned in a stratified fashion with 60% for training, 20% for validation and 20% for testing datasets and we use a batch size of 100 sequences. The hidden size in the base rule model is set to the double of the input size of the AND layer (which is the window size of the convolution). More details on experimental setting can be found in Appendix E. At each epoch (200 in total), we evaluate the model against the validation dataset and keep the model with the highest accuracy and in case of equality the model with lowest penalty. For each experiment, we run the algorithm 10 times with different weights initializations. Resulting metrics are averaged over these runs.

We run the experiments with two different window sizes (3 and 6) for the CR2N convolution filter size. We compare the two versions of the architecture: the local and global models described in Section 4 and study three different dynamic pruning strategies: none, dynamic enforced sparsity from epoch=0 and from epoch=30 (arbitrary).

# 6 RESULTS

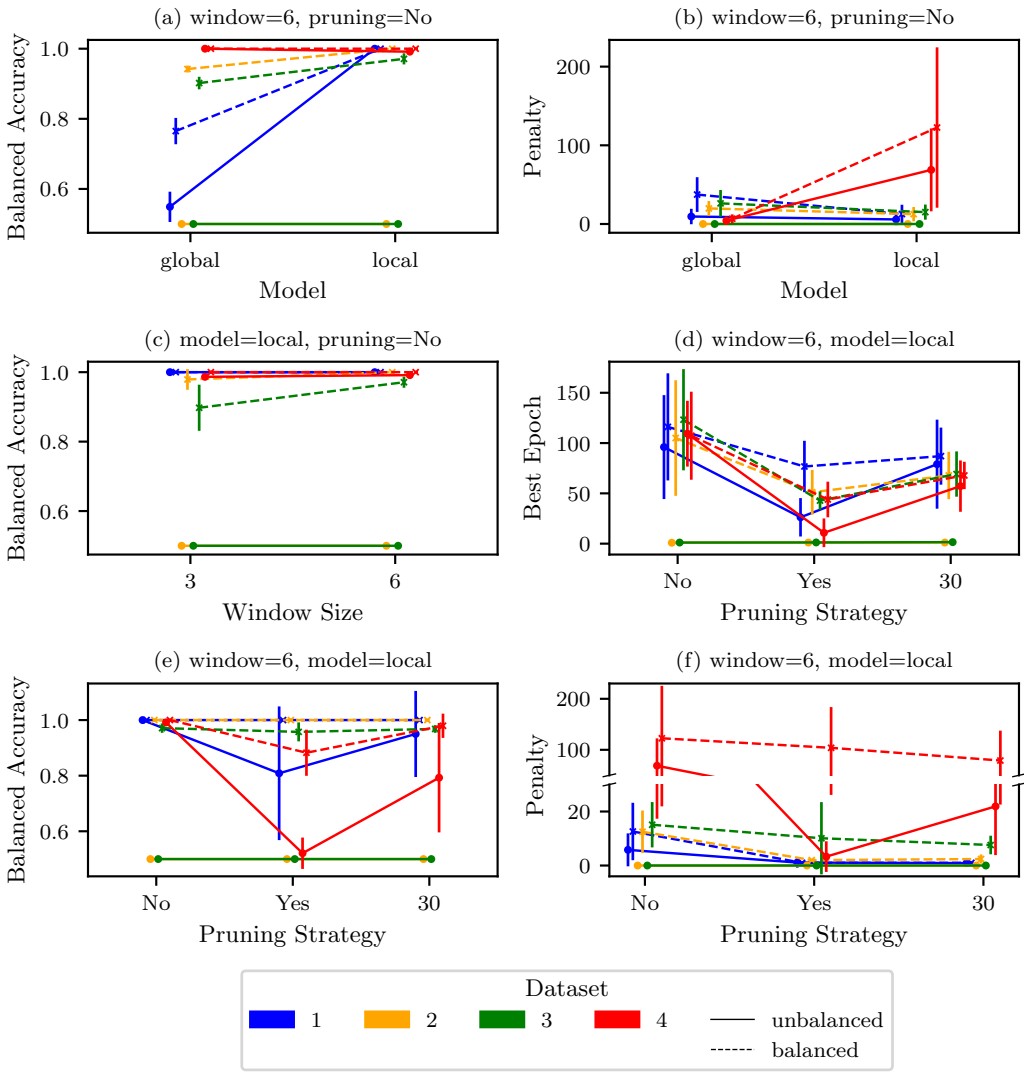

Figure 3: Representations of key results obtained on the synthethic datasets. Error bars represent the standard deviation over the 10 executions with different weights initializations. Full results are available in Appendix F.

Table 2: Performance metrics obtained for the different models, window size and pruning strategy on the peptides dataset, along with the standard deviations over the 10 executions with different weights initializations. (Bal. Acc.: balanced accuracy, Epoch: best epoch).

| Model | Window | Pruning | Accuracy | Bal. Acc. | Penalty ($\Pi$) | Epoch |
|-------|--------|---------|----------|-----------|-----------------|-------|
| global | 3 | No | $88.9 \pm 5.0$ | $75.3 \pm 12.7$ | $58.7 \pm 35.9$ | $105 \pm 60$ |
| | | Yes | $85.3 \pm 5.3$ | $67.3 \pm 14.3$ | $19.4 \pm 15.9$ | $23 \pm 20$ |
| | | 30 | $88.7 \pm 5.0$ | $75.4 \pm 12.9$ | $46.9 \pm 28.0$ | $48 \pm 22$ |
| | 6 | No | $\mathbf{91.2} \pm 1.0$ | $\mathbf{81.8} \pm 2.0$ | $220.0 \pm 56.5$ | $92 \pm 82$ |
| | | Yes | $89.5 \pm 3.6$ | $77.6 \pm 9.3$ | $132.7 \pm 93.9$ | $33 \pm 15$ |
| | | 30 | $\mathbf{91.0} \pm 1.3$ | $\mathbf{82.0} \pm 2.8$ | $\mathbf{97.3} \pm \mathbf{60.6}$ | $71 \pm 19$ |
| local | 3 | No | $90.0 \pm 3.7$ | $78.7 \pm 9.7$ | $694.3 \pm 269.3$ | $77 \pm 48$ |
| | | Yes | $\mathbf{90.3} \pm 1.5$ | $\mathbf{81.6} \pm 1.6$ | $\mathbf{885.2} \pm \mathbf{328.7}$ | $25 \pm 8$ |
| | | 30 | $89.2 \pm 5.2$ | $76.3 \pm 13.2$ | $674.7 \pm 483.9$ | $49 \pm 13$ |
| | 6 | No | $\mathbf{92.1} \pm 1.0$ | $\mathbf{83.5} \pm 2.5$ | $\mathbf{3.9k} \pm \mathbf{1.4k}$ | $91 \pm 58$ |
| | | Yes | $88.0 \pm 4.7$ | $74.6 \pm 12.4$ | $1.0k \pm 1.0k$ | $34 \pm 16$ |
| | | 30 | $\mathbf{91.7} \pm 1.5$ | $\mathbf{83.5} \pm 2.6$ | $1.8k \pm 0.6k$ | $49 \pm 10$ |

**Rule grammar and expressivity**   The importance of the rule model expressivity can be seen concretely by comparing the different patterns the local and global models have learned for dataset 3b for example: 1. (A or B)-*-(C)-*-*-(A or B or C or D or E or F) in sequence (global, no pruning, window size=6), and 2. (B at t-5 and C at t-3) or (A at t-6 and C at t-4) (local, pruning, window size=6). In the first case, the grammar is not appropriate to model the data (as a reminder, the global model is a constrained version of the local model) as opposed to the local model that learned the perfect rule. In practice, on real data, obtained patterns, such as "if (D or E or G or H or I or N or Q or T or Y)-*-*-*-(D or E or G or I or N or Q or S or T or V or Y) in sequence" obtained for labelling 'invalid-virtual' peptides, can be explored further by a domain expert. Black box approaches do not provide such insights.

It is also highlighted by comparing experiments with local or global model and experiments with different window sizes. First of all, the accuracy of the local model is higher compared to the global model on balanced synthetic datasets 1, 2 and 3 (Fig 3(a)). For balanced and unbalanced dataset 4, both models achieve very high accuracies ($> 95\%$). However, as shown on Fig 3(b), it is at the cost of rule complexity for the local approach with averaged penalty values higher than 60 (and standard deviation higher than 50) compared to lower than 10 for the global model (and standard deviation lower than 5). It points out that the local model in that case requires on average at least more than 6 times more terminal conditions in the learned rule than the global model for comparable accuracies, but also that the weights initial states have a huge impact on the rule complexity when the rule grammar is not expressive enough (with no pruning). Those results are confirmed on real-world dataset with the peptides dataset, accuracies between the local and global models especially for a window size of 6 are comparable. However there is an order of magnitude difference for the penalty, global approach being more concise. It is important to note that by architecture the global approach has less weights to train and thus a much lower maximum penalty.

Datasets 2b and 3b benefit from a bigger window size (highly expected for dataset 3/3b due to ground truth pattern size) as shown in Fig 3(c). Accuracies are also higher with window size 6 than 3 for the peptides dataset at the cost of also higher penalties (Table 2).

The more expressive the model is i.e. the more patterns it can model, training limitations aside, the better for the performance. Of course any black box neural network with no such 1-1 rule mapping constrained architecture would reach 100% accuracy, but it is that mapping in particular that makes the model relevant, expressive and fully interpretable. Also, the best performances in accuracy for the peptides dataset ($\sim 91\%$) are comparable to the best results ($\sim 92\%$) obtained from classification with single kernels when applied to that same dataset in Nwegbu et al. (2022), our model providing an additional fully-interpretable property. The presented model is also flexible due to its logical equivalence and can be inputted into other logical layers for deeper architectures to extend the rule grammar (Beck & Fürnkranz, 2021). It can also be extended for timeseries, temporal aggregates or

multi-classification problems. Other rule grammar extensions can be inspired by Linear Temporal Logic domain and regular expression pattern mining (De Giacomo et al., 2022). However the more expressive the model is the more attention is required for training and rule complexity.

**Sparsity and training strategy**   The importance of the model sparsity is pointed out by the experiments with different pruning strategies. First, looking at training scenarios, both on synthethic and peptide datasets, experiments with sparsity-during-training approaches reach the best model faster on average than without (lower best epoch Fig 3(d)). Then, regarding the performance in terms of accuracy, we can differentiate two cases: balanced and unbalanced datasets. Training of unbalanced datasets is more affected by the aggressive dynamic pruning strategy than balanced datasets with a drop in average of around 0.2 in accuracy for dataset 1 compared to an equivalent accuracy for balanced dataset 1 for example (Fig 3(e)). The pruning strategy starting after 30 epochs is preferred in both cases. Average accuracies with a pruning strategy not starting immediately (30 epochs) are comparable to the ones obtain without pruning for balanced datasets. In terms of rule complexity, penalty values are lower with pruning and even lower when starting after 30 epochs in most cases (Fig 3(f)).

With our pruning strategy (Eq 10), we make the assumption that lower positive $loc$ values are associated to overfitting or redundancy by taking into account that values closer to 0, i.e. on the sigmoid slope, are more likely to shift thus less 'certain'. As pointed out in early work by Prechelt (1997), the dynamic pruning strategy helps to overcome possible lower generalization ability compared to a fixed pruning which could explain cases of better performance (peptide dataset local model window size of 3 for example). Prechelt proposed a different pruning strategy based on a generalization loss to characterize the amount of overfitting. When this strategy is relevant in more general cases and can be applied to many different networks, our strategy is tailored for minimizing positive trainable parameter values.

Sparsity of the model is also induced via the regularization term $\Pi$ in the loss function $\mathcal{L}$ (Eq 7). While this method is parameterized with a relative importance of sparsity for training optimization and provides an uncontrolled target sparsity, a dynamic pruning strategy is easier to control for both target sparsity and accuracy but is highly dependent on the pruning schedule (Hoefler et al., 2021).

An interesting point is made by Hoefler et al. (2021) about the convolutional operator that 'can be seen as a sparse version of fully-connected layers'. That level of forced sparsity in our model is therefore defined by the fixed window size model parameter with respect to the maximum sequence length. The ideal sparser window size would be the size of the maximum temporal hidden pattern in distribution that can only be approximated with external or expert knowledge and/or tuned with trial and error.

With or without a dynamic pruning strategy, for highly unbalanced datasets (2 and 3), experiments have shown that the training strategy of the model is not suitable. Indeed most of them, label everything with the majority class (50% balanced accuracy). It corresponds to the specific case of learning an empty rule (penalty=0) (Fig 3(a,c,e)). For unbalanced datasets 1 and 4, their best models do not reach on average the same accuracies as in their balanced versions.

Overall this training strategy is both the key and the main limitation of our approach: it can provide a sharp concise rule with minimal redundancy and simplified logical expression but it is highly dependent on numerous model, training and pruning parameters and is not suited as is for highly unbalanced datasets.

## 7   CONCLUSION

To conclude, we presented a 1D-convolutional neural architecture to discover local and global patterns in sequential data while learning binary classification rules. This architecture is fully differentiable, interpretable and requires sparsity that is enforced dynamically. One main limitation is its dependence to the window size and sparsity scheduler parameters. Further work will consist in integrating this block into more complex architectures to augment the expressivity of the learned rules as well as extending it for multi-classification.

AUTHOR CONTRIBUTIONS

M.C. and R.K. designed the model. R.K. encouraged M.C. to investigate the use of convolutions and supervised the findings of this work. P.B. and F.F. helped supervise the project. M.C. validated the training strategy and carried out the implementation and the experiments. M.C. wrote the manuscript. All authors provided critical feedback and helped shape the manuscript.

ACKNOWLEDGMENTS

We would like to thank Shubham Gupta for helpful discussions and constructive feedback as well as Yusik Kim for reviewing the manuscript. This work has been partially funded by the French government as part of project PSPC AIDA 2019-PSPC-09, in the framework of the "Programme d'Investissement d'Avenir".

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

## A    CONTEXT-FREE GRAMMAR

**Context-free grammar (Chomsky, 1956)**   A context free-grammar is a 4-tuple $G = (V_T, V_N, S, R)$ where

- $V_T$, is finite set of terminals or terminal elements in the language that form the alphabet of the language.
- $V_N$, disjoint from $V_T$, is a finite set of non terminal elements (variables) that define a sublanguage of $L$. We note $V = V_N \cup V_T$, the vocabulary of the grammar.
- $S \in V_N$, is the start symbol or variable that defines the whole sentence.
- $R$ is a finite set of rules or production rules of the form $A \to w$ with $A \in V_N$ and $w \in V^*$

In the following, we have 3 different types of terminal elements on the syntax level:

- reserved words, distinguished with the following style : reserved
- signs, such as for example $-, *, ...$
- other terminal elements that are defined prior to the grammar, distinguished with the following style: terminal

Production rules presented in the paper (Eq 3, Eq 4, Eq 5 and Eq 6) define grammars when associated to values for $V_T$, $V_N$ and $S$.

Here is an example for the base rule model grammar with production rules in Eq 3.

$$
\begin{aligned}
V_T &= \{\underline{\text{if}}, \underline{\text{then class = 1 else class = 0}}, \wedge, \vee, =\} \cup \{\underline{x_1}, \underline{x_2}, \dots\} \cup \\
&\quad \left\{ \underline{x_{c_1} = \alpha_0^1}, \dots, \underline{x_{c_1} = \alpha_{n_1}^1}, \dots, \underline{x_{c_k} = \alpha_{n_k}^k} \right\} \\
V_N &= \{\text{rule, base expression, conjunction, predicate, categorical expression,} \\
&\quad \text{categorical literal, literal}\} \\
S &= \{\text{rule}\}
\end{aligned}
\tag{11}
$$

## B    HARD CONCRETE DISTRIBUTION (LOUIZOS ET AL., 2018)

Parameters are set as follows: $\beta = 2/3$, $\zeta = 1.1$ and $\gamma = -0.1$.

$$
u \sim U(0,1), \quad s = \sigma((log(u) - log(1-u) + loc)/\beta), \quad \hat{s} = s * (\zeta - \gamma) + \gamma
\tag{12}
$$

$$
W = \min(\max(\hat{s}, 0), 1)
\tag{13}
$$

## C    PEPTIDES DATASET

UCI anticancer peptides dataset (Grisoni et al., 2019) (Available on Dua & Graff (2017)) is composed of one-letter amino acid sequences (of variable length) and each sequence is labeled with its anticancer activity on breast cancer cell lines. The dataset provides 4 classes with the following distribution: 83 inactive-exp, 750 inactive-virtual, 98 moderately active and 18 very active. Sequences

lengths range from 5 to 38 letters (Mean: $17 \pm 5.5$). We transform this multi-classification problem into a binary classification problem (as done in Nwegbu et al. (2022)). Class 'inactive-virtual' is the positive class (750) and all the other are combined as the negative class (199). No other processing of the data is necessary and we leave it as is.

## D    SYNTHETIC DATASETS GENERATION

Balanced datasets are generated randomly with the same ground truth as unbalanced datasets. Then, they are upsampled until the minority class represents half of the goal dataset size and appropriate number of majority class are randomly removed.

## E    EXPERIMENTAL SETTING

The $loc$-parameters for weights computation are initialized with xavier uniform initialization method (Glorot & Bengio, 2010). The loss function is described in Eq 7 and depends on the MSE loss and regularization coefficient $\lambda = 10^{-5}$. The adam optimizer is used with a fixed learning rate set to $0.1$ and a run consists of 200 epochs.

Experiments were run on CPU on a MacBookPro18,2 (2021) with Apple M1 Max chip, 10 Cores, 32 GB of RAM and running macOS Monterey Version 12.4.

## F    FULL EXPERIMENT RESULTS

Table 3: Performance metrics obtained for the different models, window size and pruning strategy on the synthetic datasets. Values are followed by the standard deviation over the 10 executions with different weights initializations. (Bal Acc: balanced accuracy, Epoch: best epoch).

| Dataset | Model | Window | Pruning | Bal. Acc. | Penalty | Epoch |
|---|---|---|---|---|---|---|
| 1 | global | 3 | No | $50.0 \pm 0.0$ | $0.0 \pm 0.0$ | $3 \pm 1$ |
| | | | Yes | $50.0 \pm 0.0$ | $0.0 \pm 0.0$ | $3 \pm 2$ |
| | | | 30 | $50.0 \pm 0.0$ | $0.0 \pm 0.0$ | $2 \pm 1$ |
| | | 6 | No | $54.9 \pm 3.9$ | $9.5 \pm 8.1$ | $109 \pm 64$ |
| | | | Yes | $50.0 \pm 0.0$ | $0.0 \pm 0.0$ | $1 \pm 1$ |
| | | | 30 | $50.0 \pm 0.0$ | $0.0 \pm 0.0$ | $1 \pm 1$ |
| | local | 3 | No | $100.0 \pm 0.0$ | $1.8 \pm 1.5$ | $63 \pm 24$ |
| | | | Yes | $90.0 \pm 20.0$ | $0.8 \pm 0.4$ | $56 \pm 47$ |
| | | | 30 | $85.0 \pm 22.9$ | $0.7 \pm 0.5$ | $41 \pm 27$ |
| | | 6 | No | $100.0 \pm 0.0$ | $5.8 \pm 5.6$ | $96 \pm 50$ |
| | | | Yes | $80.9 \pm 23.6$ | $1.1 \pm 1.4$ | $26 \pm 18$ |
| | | | 30 | $95.0 \pm 15.0$ | $0.9 \pm 0.3$ | $79 \pm 43$ |
| 1b | global | 3 | No | $70.7 \pm 4.0$ | $5.1 \pm 1.6$ | $81 \pm 27$ |
| | | | Yes | $68.5 \pm 6.6$ | $5.8 \pm 2.8$ | $47 \pm 31$ |
| | | | 30 | $71.6 \pm 3.5$ | $5.0 \pm 1.3$ | $60 \pm 16$ |
| | | 6 | No | $76.5 \pm 3.4$ | $37.4 \pm 20.3$ | $108 \pm 49$ |
| | | | Yes | $70.2 \pm 4.3$ | $12.5 \pm 5.5$ | $29 \pm 15$ |
| | | | 30 | $72.7 \pm 5.1$ | $15.4 \pm 6.9$ | $53 \pm 13$ |
| | local | 3 | No | $100.0 \pm 0.0$ | $3.0 \pm 3.1$ | $70 \pm 47$ |
| | | | Yes | $100.0 \pm 0.0$ | $1.0 \pm 0.0$ | $49 \pm 39$ |
| | | | 30 | $100.0 \pm 0.0$ | $1.0 \pm 0.0$ | $54 \pm 30$ |
| | | 6 | No | $100.0 \pm 0.0$ | $12.6 \pm 10.1$ | $116 \pm 52$ |
| | | | Yes | $100.0 \pm 0.0$ | $1.0 \pm 0.0$ | $77 \pm 24$ |
| | | | 30 | $100.0 \pm 0.0$ | $1.0 \pm 0.0$ | $87 \pm 27$ |

**Table 3 continued from previous page**

| Dataset | Model | Window | Pruning | Bal. Acc. | Penalty | Epoch |
|---|---|---|---|---|---|---|
| 2 | global | 3 | No | $50.0 \pm 0.0$ | $0.0 \pm 0.0$ | $3 \pm 1$ |
| | | | Yes | $50.0 \pm 0.0$ | $0.0 \pm 0.0$ | $3 \pm 1$ |
| | | | 30 | $50.0 \pm 0.0$ | $0.0 \pm 0.0$ | $2 \pm 1$ |
| | | 6 | No | $50.0 \pm 0.0$ | $0.0 \pm 0.0$ | $1 \pm 1$ |
| | | | Yes | $50.0 \pm 0.0$ | $0.0 \pm 0.0$ | $1 \pm 0$ |
| | | | 30 | $50.0 \pm 0.0$ | $0.0 \pm 0.0$ | $1 \pm 1$ |
| | local | 3 | No | $50.0 \pm 0.0$ | $0.0 \pm 0.0$ | $1 \pm 1$ |
| | | | Yes | $50.0 \pm 0.0$ | $0.0 \pm 0.0$ | $1 \pm 1$ |
| | | | 30 | $50.0 \pm 0.0$ | $0.0 \pm 0.0$ | $1 \pm 1$ |
| | | 6 | No | $50.0 \pm 0.0$ | $0.0 \pm 0.0$ | $1 \pm 1$ |
| | | | Yes | $50.0 \pm 0.0$ | $0.0 \pm 0.0$ | $1 \pm 0$ |
| | | | 30 | $50.0 \pm 0.0$ | $0.0 \pm 0.0$ | $1 \pm 0$ |
| 2b | global | 3 | No | $90.5 \pm 0.0$ | $3.7 \pm 2.1$ | $44 \pm 37$ |
| | | | Yes | $85.7 \pm 9.8$ | $2.3 \pm 0.6$ | $39 \pm 21$ |
| | | | 30 | $90.5 \pm 0.0$ | $2.0 \pm 0.0$ | $51 \pm 24$ |
| | | 6 | No | $94.2 \pm 0.6$ | $19.8 \pm 7.7$ | $89 \pm 36$ |
| | | | Yes | $90.5 \pm 0.0$ | $2.0 \pm 0.0$ | $71 \pm 16$ |
| | | | 30 | $91.0 \pm 1.5$ | $12.4 \pm 13.0$ | $61 \pm 26$ |
| | local | 3 | No | $97.9 \pm 2.6$ | $2.8 \pm 1.2$ | $58 \pm 42$ |
| | | | Yes | $97.4 \pm 2.6$ | $1.5 \pm 0.5$ | $43 \pm 28$ |
| | | | 30 | $98.0 \pm 2.4$ | $1.6 \pm 0.5$ | $54 \pm 29$ |
| | | 6 | No | $100.0 \pm 0.0$ | $12.5 \pm 7.4$ | $105 \pm 56$ |
| | | | Yes | $100.0 \pm 0.0$ | $2.0 \pm 0.0$ | $51 \pm 21$ |
| | | | 30 | $100.0 \pm 0.0$ | $2.4 \pm 1.2$ | $68 \pm 22$ |
| 3 | global | 3 | No | $50.0 \pm 0.0$ | $0.0 \pm 0.0$ | $2 \pm 1$ |
| | | | Yes | $50.0 \pm 0.0$ | $0.0 \pm 0.0$ | $2 \pm 1$ |
| | | | 30 | $50.0 \pm 0.0$ | $0.0 \pm 0.0$ | $2 \pm 1$ |
| | | 6 | No | $50.0 \pm 0.0$ | $0.0 \pm 0.0$ | $1 \pm 1$ |
| | | | Yes | $50.0 \pm 0.0$ | $0.0 \pm 0.0$ | $1 \pm 0$ |
| | | | 30 | $50.0 \pm 0.0$ | $0.0 \pm 0.0$ | $1 \pm 0$ |
| | local | 3 | No | $50.0 \pm 0.0$ | $0.0 \pm 0.0$ | $1 \pm 1$ |
| | | | Yes | $50.0 \pm 0.0$ | $0.0 \pm 0.0$ | $2 \pm 1$ |
| | | | 30 | $50.0 \pm 0.0$ | $0.0 \pm 0.0$ | $1 \pm 1$ |
| | | 6 | No | $50.0 \pm 0.0$ | $0.0 \pm 0.0$ | $1 \pm 0$ |
| | | | Yes | $50.0 \pm 0.0$ | $0.0 \pm 0.0$ | $1 \pm 1$ |
| | | | 30 | $50.0 \pm 0.0$ | $0.0 \pm 0.0$ | $2 \pm 1$ |
| 3b | global | 3 | No | $79.8 \pm 8.1$ | $4.6 \pm 2.2$ | $98 \pm 51$ |
| | | | Yes | $70.5 \pm 8.1$ | $3.1 \pm 1.4$ | $48 \pm 31$ |
| | | | 30 | $80.4 \pm 9.2$ | $2.8 \pm 0.6$ | $57 \pm 36$ |
| | | 6 | No | $90.2 \pm 1.4$ | $26.2 \pm 15.3$ | $120 \pm 51$ |
| | | | Yes | $77.4 \pm 7.4$ | $14.1 \pm 8.0$ | $35 \pm 13$ |
| | | | 30 | $87.7 \pm 3.1$ | $11.5 \pm 6.6$ | $61 \pm 10$ |
| | local | 3 | No | $89.8 \pm 6.3$ | $8.5 \pm 3.9$ | $95 \pm 64$ |
| | | | Yes | $86.2 \pm 8.0$ | $6.1 \pm 8.2$ | $50 \pm 34$ |
| | | | 30 | $92.5 \pm 4.6$ | $4.2 \pm 1.8$ | $64 \pm 22$ |
| | | 6 | No | $97.1 \pm 1.2$ | $15.1 \pm 7.9$ | $123 \pm 49$ |
| | | | Yes | $95.7 \pm 2.9$ | $10.1 \pm 12.9$ | $43 \pm 8$ |
| | | | 30 | $96.8 \pm 0.9$ | $7.6 \pm 2.9$ | $69 \pm 21$ |

**Table 3 continued from previous page**

| Dataset | Model | Window | Pruning | Bal. Acc. | Penalty | Epoch |
|---------|-------|--------|---------|-----------|---------|-------|
| 4 | global | 3 | No | $100.0 \pm 0.0$ | $2.2 \pm 0.6$ | $42 \pm 28$ |
| | | | Yes | $90.0 \pm 20.0$ | $1.6 \pm 0.8$ | $28 \pm 14$ |
| | | | 30 | $95.0 \pm 15.0$ | $1.8 \pm 0.6$ | $42 \pm 25$ |
| | | 6 | No | $100.0 \pm 0.0$ | $4.3 \pm 3.4$ | $72 \pm 35$ |
| | | | Yes | $86.2 \pm 21.3$ | $1.7 \pm 0.9$ | $33 \pm 20$ |
| | | | 30 | $100.0 \pm 0.0$ | $2.0 \pm 0.0$ | $43 \pm 15$ |
| | local | 3 | No | $98.7 \pm 0.4$ | $28.6 \pm 14.5$ | $108 \pm 40$ |
| | | | Yes | $59.1 \pm 12.9$ | $3.4 \pm 5.1$ | $30 \pm 37$ |
| | | | 30 | $77.1 \pm 20.7$ | $12.5 \pm 8.7$ | $58 \pm 39$ |
| | | 6 | No | $99.1 \pm 0.6$ | $68.8 \pm 51.0$ | $109 \pm 31$ |
| | | | Yes | $52.1 \pm 5.2$ | $3.3 \pm 5.2$ | $11 \pm 13$ |
| | | | 30 | $79.3 \pm 19.2$ | $21.9 \pm 17.5$ | $57 \pm 24$ |
| 4b | global | 3 | No | $100.0 \pm 0.0$ | $2.7 \pm 1.1$ | $16 \pm 8$ |
| | | | Yes | $99.2 \pm 2.4$ | $2.1 \pm 0.3$ | $29 \pm 18$ |
| | | | 30 | $100.0 \pm 0.0$ | $2.0 \pm 0.0$ | $42 \pm 32$ |
| | | 6 | No | $100.0 \pm 0.0$ | $6.7 \pm 3.8$ | $90 \pm 52$ |
| | | | Yes | $99.2 \pm 2.4$ | $4.7 \pm 8.1$ | $50 \pm 22$ |
| | | | 30 | $100.0 \pm 0.0$ | $2.2 \pm 0.6$ | $69 \pm 25$ |
| | local | 3 | No | $100.0 \pm 0.0$ | $39.9 \pm 17.5$ | $76 \pm 38$ |
| | | | Yes | $99.0 \pm 2.0$ | $29.2 \pm 9.4$ | $60 \pm 14$ |
| | | | 30 | $98.7 \pm 4.0$ | $35.9 \pm 17.3$ | $53 \pm 14$ |
| | | 6 | No | $100.0 \pm 0.0$ | $122.6 \pm 100.2$ | $107 \pm 42$ |
| | | | Yes | $88.2 \pm 7.8$ | $103.9 \pm 77.3$ | $44 \pm 16$ |
| | | | 30 | $98.0 \pm 3.9$ | $79.0 \pm 55.9$ | $68 \pm 12$ |

