# OpenReview forum: "Neural-based classification rule learning for sequential data"
_ICLR.cc/2023/Conference — ICLR 2023 poster_

### Official Review · Reviewer_r7kZ · 2022-10-18

**Confidence:** 3
**Clarity, Quality, Novelty And Reproducibility:** The paper is clearly written and some…
**Correctness:** 3
**Technical Novelty And Significance:** 3
**Empirical Novelty And Significance:** 3
**Recommendation:** 6

**Strength And Weaknesses:**

Strengths. The paper introduces a method that can be useful for real applications. The paper is well-written.

Weaknesses. Comparison to state-of-the-art methods are missing.
The local and global patters are not so easy to understand. In general, in my opinion, Section 2.2 is not easy to follow.

Two window values (for sequences), namely, 3 and 6 were tested. It is difficult to make strong conclusions from results based on these two values only. It is also mentioned that it is a good idea to take a window size which equals to the maximum length. First, I wonder whether if such a big window is taken, the model would suffer from overfitting. Second, it is indeed interesting to take quite a big window but also include into the model some shorter windows (similarly to n-gram models). Third, windows of various lengths (performing data segmentation) would be an optimal choice.

Speaking about interpretability: what kind of rules were learnt?
I did not really see that the interpretable aspect of the method was validated, i.e., that some rules were extracted and considered/validated by human experts.

Other remarks. In Section 3, "the model is trained via automatic differentiation", and it is also mentioned that the losses are differentiable. Why do you prefer automatic differentiation if the first derivative can be computed?

In Section 3, the paragraph "Loss function" is not very easy to understand: why there is product of three \Pi matrices? Is the multiplication coordinate-wise, if yes, do they have the same dimensions? What is \Pi_{stack} and why the sum is taken over \Pi_{stack} matrices?

It is mentioned that both balanced and unbalanced data sets are considered. How the data sets are balanced? I did not find any information on whether some undersampling or weighted loss was used.

In Table 2: what the column "Penalty" contains?

**Summary Of The Paper:**

The paper proposes a method based on a binary neural network to learn classification rules from sequential data. The model contains a stackedOR layer for categorical variables, a logical AND layer and an OR layer. The training strategy includes dynamically-induced sparsity.


**Summary Of The Review:**

The method is sound but some points should be clarified.

---

> ### Author Response · Authors · 2022-11-12
> **Response to Reviewer r7kZ**
>
> Thank you very much for your review. Your comments have been addressed below. The manuscript has been updated (check the yellow highlights) to reflect all the changes promised below.
>
> **Regarding local and global patterns:**
>
> A local pattern occurs at a specific location in the sequence. For example, it can classify a sentence as a negative sentence if "the first word is 'never'" and "the second word is a verb" or:
>
> if ("Never" at position 1) AND ("Do" OR "Have" OR "Play" or "Walk" at position 2)
>
> A global pattern can occur anywhere in the sequence. For example, it can classify a sequence of bank transactions as suspicious if "there are at least 3 consecutive failed transactions in the sequence" or:
>
> (Fail-Fail-Fail) in the sequence.
>
> We have clarified this further in the manuscript.
>
> **Comparison to state-of-the-art methods:**
>
> To our knowledge, there is no comparable state-of-the-art method that can learn rule-based global/local patterns in sequential data like our model. As exemplified in the previous point, being able to learn such patterns is useful in practice. In the absence of a comparable method, we can either compare with black box methods like neural networks or with ad-hoc adaptations of simple rule learning methods like RIPPER for example.
>
> A comparison with black box methods would be unfair because of our focus on interpretability and succinctness of rules, which often restricts the complexity of the learned models. Nevertheless, as our experiments with the peptides data shows, our model does produce reasonably accurate rules.
>
> Concerning simple rule learning methods, one can in principle pass sequences as a single input vector to a rule learning algorithm like RIPPER. However, to represent a global pattern such as "A-B in sequence", RIPPER would learn a DNF of the form:
>
> ($X_1=A$ AND $X_2=B$)   OR  ($X_2=A$ AND $X_3=B$)   OR   ….   OR  ($X_{n-1}=A$ AND $X_n=B$)
>
> This is because the grammar of rules learned by RIPPER (and other rule learning methods) is not expressive enough to succinctly represent a global pattern.
>
> We therefore focussed on the new dimension of local vs global patterns that are learned only by our model and chose not to compare with other baselines.
>
> **Regarding various window sizes and overfitting:**
>
> The size of the convolution window determines the complexity of the pattern learned by the base model. To learn a pattern involving inputs $x_t$ and $x_{t + k}$, we need a convolutional window of size at least $k$.
>
> First, we want to clarify that our recommendation for the ideal window size was based on the (unknown) maximum temporal width in the ground truth pattern ($k$ in the example above) and not the maximum length of the sequence.
>
> Second, we wish to emphasize that the window size is a hyper-parameter and one must use domain knowledge and trial-and-error to find a good value for it, as in a standard CNN.
>
> Third, bigger windows allow learning more complex patterns that may indeed lead to overfitting. Tuning the window-size hyper parameter avoids this issue (again, as in standard CNNs).
>
> Finally, it is indeed possible to extend our model to use multiple convolutional filters (possibly of different sizes). We leave this for future work. Thanks for the suggestion.
>
> **Validating the interpretability of learned rules:**
>
> Please see our third comment in response to Reviewer UfDD regarding the interpretability of rules on a real-world dataset.
>
> We agree with you that a human expert’s opinion about the rules would be invaluable, but we believe that it is outside the scope of our paper. We take the following as established facts: 1) rules are interpretable and 2) shorter rules are better. This is confirmed by several studies (Miller, 2017) and a push towards widespread adoption of rules in the industry (Rudin, 2019). While it is interesting to consider alternative ways to quantify the interpretability of rules, our focus is on the algorithmic problem of finding rules given a certain notion of their interpretability (namely, their length).
>
> **Why use automatic differentiation?**
>
> As is the case with with most neural networks, while computing derivates analytically is in principle possible, it is often very cumbersome. This is why people use efficient automatic differentiation frameworks like PyTorch to make implementations easy. Please also see our response to Reviewer JYCZ  for more information on the differentiability of our model.
>
> **Regarding $\Pi$ matrices and penalty terms:**
>
> Concerns on the clarity of that paragraph were also raised by Reviewer JYCZ. We hope our answer there also addresses your concerns.
>
> **Regarding balanced datasets:**
>
> All ground truth rules in our synthetic data experiments lead to unbalanced datasets. To get a balanced dataset, we generate many more examples and then downsample the majority class until the classes are balanced (see Appendix A).

---

### Official Review · Reviewer_JYCZ · 2022-10-25

**Confidence:** 3
**Clarity, Quality, Novelty And Reproducibility:** The model is not described with enoug…
**Correctness:** 3
**Technical Novelty And Significance:** 3
**Empirical Novelty And Significance:** Not applicable
**Recommendation:** 6

**Strength And Weaknesses:**

**positive**

Architecture with potential for xAI

A very interesting way to learn rules with gradient descent

**negative**

n section 3, the authors should define the notation and matrix dimensions: I don't understand (6) and (7).
I can't understand the regularization procedure.

I am not familiar with logical networks, but I don't get how the author make the architecture differentiable




**Summary Of The Paper:**

Learning rules is difficult and often lead to low performances. From the computational point of view, it poses a problem of differentiability. This article propose to learn both the discriminant pattern and explicit decision rules.
The authors first insist on the cabablities associated with a 3 layer rule network. Then, they describe the convolution layer which applies rules on the discrete signal. Section 3 explains how the architecture is learnt. Then, the authors conduct experiments on synthetic and real discrete signals.

**Summary Of The Review:**

The idea of the article is really interesting but the whole model description (section 3), in particular the learning process, is impossible to understand (at least for a reviewer that is not familiar with rule neural networks). For this reason, I can't accept this article.


**AFTER REBUTAL**

The paper has been significantly improved by the authors. The procedure is much more detailed than in the original version and an anonymous repository has been made available. This encourage me to change my ratings.

I am not familiar enough with the previous contribution in the field to assess reliably the novelty of the proposal.

---

> ### Author Response · Authors · 2022-11-12
> **Response to Reviewer JYCZ**
>
> Thank you very much for your review. Your comments have been addressed below. The manuscript has been updated (check the yellow highlights) to reflect all the changes promised below.
>
> **Concerning the penalty terms and the regularisation procedure:**
>
> Intuitively, $\Pi_{local}$ and $\Pi_{global}$ count the number of terminal conditions in the learned rules for the local and global model respectively. These quantities depend on $\Pi_{base}$ which uses a recursive weighted sum to count the total terminal conditions. Our regularisation procedure attempts to constrain $\Pi_{local}$ or $\Pi_{global}$ depending on the model being trained. Please see our second comment to Reviewer UfDD for more intuition about the regularization procedure.
>
> We have added more details about this in the manuscript, especially concerning the dimensions of various quantities. Thanks for the suggestions. For your convenience and to aid the readability of this response, we will add a separate comment to explain the computation of $\Pi$ in detail.
>
> **Regarding the differentiability of the architecture:**
>
> We use two tricks to make our architecture differentiable. First we use binary weights that are trained indirectly via a continuous parameter which is activated by a sigmoid function (latent weights).
>
> Second, we use continuous relaxation of the logical AND and OR operations (eq. 1 and 2). This makes our loss function differentiable w.r.t the input to the sigmoid function in the first point above (see Kusters et al. (2022) for details). We have added more clarity on this to the manuscript.
>
> **The model is not described with enough details to enable reproducibility.**
>
> We have added more implementation details to the manuscript (see Section 3) and shared our code. We’ll be happy to answer more questions to improve your confidence in the reproducibility of our results.

---

> ### Author Response · Authors · 2022-11-12
> **Computation of $\Pi$**
>
> Consider a size $K$ sub-sequence $[a_1, a_2, …, a_K]$ of the given sequence where each $a_i$ can take one of n possible values from the set $V = ${$v_1, v_2, …, v_n$}. Intuitively, a StackedOR layer maps each $a_j$ to a binary value $x_j$ which evaluates to 1 if $a_j$ belongs to a subset $V_j$ of $V$. The learned binary weight $W_{stack}^j \in ${$0, 1$}$^n$ indicates which elements of $V$ are present in $V_j$.
>
>
> Now, $\Pi_{stack}$ is a $K$-dimensional vector where the $j^{th}$ entry corresponds to the number of non-zero weights in $W_{stack}^j$, i.e., the number of input possibilities $|V_j|$ at position $j$ that activate $x_j$. The simpler the rules, the smaller the entries in $\Pi_{stack}$ will be.
>
>
> The inner sum in $\Pi_{base}$ adds the $\Pi_{stack}$ entries for $x_j$’s that are used by each conjunction, thereby computing the total number of elements from set $V$ that appear in each conjunction.
>
> The outer sum adds the inner sum values for the conjunctions chosen by the base model DNF. This gives the total number of elements from set $V$ that appear in the DNF. The larger this number, the more complicated the DNF is.
>
>
> $\Pi_{global}$ is equal to $\Pi_{base}$ because the weights of ConvOR layer are set to 1 in the global model.
>
> $\Pi_{local}$ multiplies $\Pi_{base}$ with the number of non-zero weights in the ConvOR layer. A concise local pattern will have very few non-zero weights.
>
> $\Pi^*$ is the ideal case scenario. Ideally, we would like to switch to the penalty for local patterns if only a few ConvOR layer weights are one and to the penalty for global patterns if many ConvOR layer weights are one. However, this is discontinuous and hence non-differentiable.
>
> Instead, we learn local and global models separately constraining the appropriate $\Pi_{local}$ or $\Pi_{global}$.

---

### Official Review · Reviewer_UfDD · 2022-10-25

**Confidence:** 2
**Correctness:** 3
**Technical Novelty And Significance:** 3
**Empirical Novelty And Significance:** 2
**Recommendation:** 8

**Clarity, Quality, Novelty And Reproducibility:**

Clarity
In general the paper is easy to follow. However, the paper could be polished in terms of clarity. For example, the authors should explain the notations for grammar in equations 3-5 for better presentation. Equation 9 is also not explained with the motivation on using such form for the rate.

Quality, Novelty And Reproducibility
The paper is novel for proposing rule learning model for sequence data. There is no code provided and detailed experiment setting so I doubt with reproducibility, which downgrades the quality of the paper. The paper could also be improved by more explanations/motivations on the techniques used and providing experiment result on explainability.

**Strength And Weaknesses:**

Strength
1. The paper improves on the base rule model by operating on sequence data and could learn global/local pattern
2. The training strategy on penalizing the complexity of the rule and sparsify-during-training is useful and could benefit other rule learning algorithm

Weaknesses
1. The proposed method does not work for unbalanced data for learning an empty rule. I am wondering if up/down-sample the positive/negative class helps
2. The paper could be polished in terms of clarity. For example, the authors should explain the notations for grammar in equations 3-5 for better presentation. Equation 9 is also not explained with the motivation on using such form for the rate.
3. There is no experiment results on explainability on real dataset. The paper will be stronger and better motivated with such a demo.

**Summary Of The Paper:**

The paper proposes a convolutional binary neural network to learn binary classification rule for sequence data. The learned rule is able to learn global/local pattern for better explainability. Furthermore, the authors propose a training strategy with latent weights and regularization term penalizing the complexity of the rule, and dynamically enforces the sparsity using sparsity-during-training method. Experiment results on 4 synthetic datasets and one real dataset are shown to exhibit the effectiveness of the proposed method.

**Summary Of The Review:**

Overall I think the paper contributes a novel method for learning rule with sequence data, and recommend for acceptance.

---

> ### Author Response · Authors · 2022-11-12
> **Response to Reviewer UfDD**
>
> Thank you very much for your review. Your comments have been addressed below. The manuscript has been updated (check the yellow highlights) to reflect all the changes promised below.
>
> **Regarding empty rules on unbalanced datasets:**
>
> This is indeed true for highly imbalanced datasets (<5% positive class). As we optimized for predictive performance under a hard constraint on rule conciseness (Section 3), the empty rule is sometimes a good tradeoff for the model. Relaxing the sparsity constraint may help; so does up- or down-sampling Quick experiments were done with down sampling following your comment on the most unbalanced dataset (#2) to validate this improvement.
>
> With the local model, a window size of 3 and pruning, on 10 different runs we obtained the following rule lengths [3, 108, 4, 0, 2, 9, 5, 7, 1, 18] compared to only empty rules without downsampling. (Training is not great every time with so little data. Also, performance metrics metrics are improved but do not mean much on testing dataset of 6 elements).
>
> **Regarding clarity of equations 3-5 (rule grammar), equation 9 (regularisation), and our experimental setup:**
>
> Thanks for the suggestions. We have expanded the description of the rule grammar (eq. 3-5) and added a section on context-free grammars (Appendix A). Equation 9 (now 10) prunes weights for which the |loc| value is not large enough. We are making the assumption that a pruning based on weight intensity values |loc| is helping the model to generalize. The pruning rate r was introduced by Zhu and Gupta (2017) and their intuition “is to prune the network rapidly in the initial phase when the redundant connections are abundant and gradually reduce the number of weights being pruned each time as there are fewer and fewer weights remaining in the network”.
>
> This has been clarified in the revision. We have also added more details about our experimental setup in the Appendix.
>
> **Regarding explainability on real dataset:**
>
> Rules are widely accepted to be interpretable with rule length being a common proxy for their interpretability. In practice, rules show patterns that can be explored further by a domain expert. For example, we learn the following rule on the peptide dataset:
>
> > if (D OR E OR G OR H OR I OR N OR Q OR T OR Y)-*-*-*-(D OR E OR G OR I OR N OR Q OR S OR T OR V OR Y) in sequence then class = ‘invalid-virtual’
>
> The letters are amino acids and * represents “any letter”. A researcher can now try to understand why sequences of five amino acids starting and ending with the letters above are interesting. Black box models do not provide such insights. We have added more clarity in the paper (Section 6).
>
> Note also that we demonstrate that our method recovers correct ground truth rules on synthetic datasets and our sparsity penalty encourages learning concise rules (Section 6).
>
> **Regarding code:**
>
> We are in the process of an internal IP clearance and will release the code (along with the datasets) publicly with the camera ready version. Until then, we have shared the code with the reviewers via a private comment.

---

### Decision · Program_Chairs · 2023-01-20

**Decision:**

Accept: poster

**Justification For Why Not Higher Score:**

To warrant a higher score, the experimental evaluation should have been more extensive.

**Justification For Why Not Lower Score:**

The approach has several novel aspects.

**Metareview: Summary, Strengths And Weaknesses:**

The paper presents a new method for interpretable classification of sequential data using a binary CNN with an interpretable filter, trained using dynamically-enforced sparsity. The method has several novel aspects - the global/local pattern learning and using sparsity during training and rule learning with gradient descent. The main weakness of the paper was its clarity. However, this aspect was improved considerably following the rebuttal stages. Reviewer r7kZ has correctly pointed out that the paper does not include comparisons with existing work, however, given that most other work in XAI does not perform rule learning, an appropriate baseline is not immediately evident. Because this paper essentially comes up with a model that other deep XAI methods do not, and because it has several novel aspects, I conclude that it would be a useful paper to include in the ICLR proceedings.

**Note From Pc:**

if the above contains the word "oral" or "spotlight" please see: "oral" presentation means -> notable-top-5% and "spotlight" means -> notable-top-25%. As stated in our emails, we are disassociating presentation type from AC recommendations

**Summary Of Ac-Reviewer Meeting:**

N/A